# Back to the basics: Clinical assessment yields robust mortality prediction and increased feasibility in low resource settings

**Mark T. Yost**[1]◎*, **Melissa M. Carvalho**[1]◎, **Lidwine Mbuh**[2‡], **Fanny N. Dissak-Delon**[3‡], **Rasheedat Oke**[1]◎, **Debora Guidam**[2‡], **Rene M. Nlong**[2‡], **Mbengawoh M. Zikirou**[2‡], **David Mekolo**[2‡], **Louis H. Banaken**[2‡], **Catherine Juillard**[1‡], **Alain Chichom-Mefire**[2‡], **S. Ariane Christie**[1]◎

1 Department of Surgery, Program for the Advancement of Surgical Equity, University of California Los Angeles, Los Angeles, California, United States of America, 2 Faculty of Health Sciences, University of Buea, Buea, Cameroon, 3 Littoral Regional Delegation, Ministry of Public Health, Yaoundé, Cameroon

◎ These authors contributed equally to this work.
‡ LM, FND-D, DG, RMN, MMZ, DM, LHB, CJ and AC-M also contributed equally to this work.
* myost@mednet.ucla.edu

**Data Availability Statement:** De-identified data Excel file uploaded as Supporting Information.

## Abstract

### Introduction

Mortality prediction aids clinical decision-making and is necessary for trauma quality improvement initiatives. Conventional injury severity scores are often not feasible in low-resource settings. We hypothesize that clinician assessment will be more feasible and have comparable discrimination of mortality compared to conventional scores in low and middle-income countries (LMICs).

### Methods

Between 2017 and 2019, injury data were collected from all injured patients as part of a prospective, four-hospital trauma registry in Cameroon. Clinicians used physical exam at presentation to assign a highest estimated abbreviated injury scale (HEAIS) for each patient. Discrimination of hospital mortality was evaluated using receiver operating characteristic curves. Discrimination of HEAIS was compared with conventional scores. Data missingness for each score was reported.

### Results

Of 9,635 presenting with injuries, there were 206 in-hospital deaths (2.2%). Compared to 97.5% of patients with HEAIS scores, only 33.2% had sufficient data to calculate a Revised Trauma Score (RTS) and 24.8% had data to calculate a Kampala Trauma Score (KTS). Data from 2,328 patients with all scores was used to compare models. Although statistically inferior to the prediction generated by RTS (AUC 0.92–0.98) and KTS (AUC 0.93–0.99), HEAIS provided excellent overall discrimination of mortality (AUC 0.84–0.92). Among 9,269 patients with HEAIS scores was strongly predictive of mortality (AUC 0.93–0.96).

**Funding:** The Cameroon Trauma Registry was supported by University of California San Francisco and University of California Los Angeles Departments of Surgery research funding (CJ) and NIH R21TW010453 grant (CJ, ACM, and SAC). This publication was supported in part by the H H LEE RESEARCH PROGRAM grant (MTY). H H LEE RESEARCH PROGRAM had no role in study design, data collection and analysis, decision to publish, or preparation of the manuscript.

**Competing interests:** The authors have declared that no competing interests exist.

## Conclusion

Clinical assessment of injury severity using HEAIS strongly predicts hospital mortality and far exceeds conventional scores in feasibility. In contexts where traditional scoring systems are not feasible, utilization of HEAIS could facilitate improved data quality and expand access to quality improvement programming.

## Introduction

Injury kills over 5 million people annually and leaves many more disabled worldwide [1]. Though about 90% of all injury deaths occur in low-to-middle income countries (LMICs), access to trauma care in these settings is far less when compared with high-income countries (HICs) and insufficiently addresses this burden [1–4]. To reduce death and disability of injury in LMICs, the World Health Organization (WHO) recommends increasing access to essential trauma care and ensuring high quality of care through outcome measurements [3, 5]. Implementation of trauma quality improvement processes has decreased injury mortality in LMICs; however, the success of sustained quality improvement depends on the ability to standardize and measure patient outcomes [6].

Injury severity scoring plays a critical role in benchmarking outcomes and developing evidenced-based trauma care systems [7]. Quantification of the severity of injury can aid patient triage, direct resource allocation, and provide risk adjustment in quality improvement (QI) analyses [7–9]. However, conventional scoring systems created and validated in HICs underperform and lose feasibility in LMICs due to resource constraints [4, 9–11]. The anatomic-based Injury Severity Score (ISS) depends on cross-sectional imaging that is often not feasible in LMICs [10]. The Revised Trauma Score (RTS) has demonstrated feasibility in LMICs by comparing weighted variables of systolic blood pressure, respiratory rate, and Glasgow coma scale (GCS) [12, 13]. Similarly, the Kampala Trauma Score (KTS), a system derived in a LMIC setting, utilizes patient age, number of serious injuries, systolic blood pressure, respiratory rate, and neurologic status to quantify injury severity [12, 14]. Nevertheless, both the RTS and KTS may lack feasibility in settings with constrained resources and perform variably in different LMICs [9, 12, 15]. For this reason, publications from LMICs may provide an estimated metric of severity, however estimation methods vary widely, and few studies have formally compared prediction capacity of these metrics with conventional scores [16, 17].

Trauma causes approximately 7% of all mortalities and contributes to significant disability in Cameroon [18, 19]. One study determined that nearly half of all emergency department (ED) visits were due to injury [17]. This finding likely underrepresents the true prevalence of injury, as a population-based survey discovered that only about 60% of injured patients seek formal care [20]. Since 2015, the Cameroon Trauma Registry (CTR) has collected detailed data on injured patients at multiple sites in Cameroon to provide a basis for ongoing trauma quality improvement [21, 22]. As in many LMICs, routine calculation of conventional anatomic injury scores such as ISS has not been widely feasible in Cameroon due to limited availability of cross-sectional imaging. As part of the CTR, the clinical care team estimates an Abbreviated Injury Scale (AIS) for each anatomic region based on physical exam findings [23]. The Highest Estimated Abbreviated Injury Scale (HEAIS) for each patient represents the clinical gestalt regarding the patient's physiologic status. In this study we evaluate the feasibility and efficacy of this metric to predict trauma mortality against previously validated metrics in a prospective, multi-center cohort. We hypothesized that the HEAIS score would be more feasible and have comparable predictive capacity compared to the previously validated scores.

## Methods

### Study design

We performed an analysis of prospective data collected between October 2017 and December 2019 as part of the multi-site national CTR. For each patient enrolled in the CTR, we calculated RTS, KTS, and HEAIS scores by using documented clinical variables. We then compared these scores to determine feasibility and discern mortality outcomes.

### Setting

Four regional hospitals in Cameroon participate in national CTR data collection [21, 22, 24]. One facility is a 710-bed public national referral hospital that serves a catchment population greater than 3 million. The second institution is a 200-bed public hospital in the Southwest region of Cameroon with a mixed rural and urban catchment of approximately 130,000 people. The third hospital is a 120-bed mission hospital in rural Littoral region along Cameroon's busiest highway, a known high-volume road traffic injury (RTI) corridor, with a catchment population of 200,000 people. The final facility is a 100-bed public hospital also along Cameroon's busiest highway, with a catchment population of greater than 100,000 people.

### Participants and data sources

We extracted data from the previously described CTR at four participating hospitals [18, 22]. All injured patients presenting to the emergency departments of the four participating hospitals between 2017 and 2019 and enrolled in the CTR were included. Trauma care team physicians used physical exam to estimate patient AIS in each anatomic region. The AIS classifies injury severity by rating each region according to a six-point scale (1 is minor, 2 is moderate, 3 is serious, 4 is severe, 5 is possibly fatal/likely to die, and 6 is fatal/currently untreatable) [23]. The AIS anatomic regions consisted of head, neck and cervical spine, face, chest and thoracic spine, abdomen, pelvis and lumbar spine, extremity, and general. Patient clinical courses were followed from ED presentation until clinical disposition. All data was collected on paper forms and subsequently transferred to an encrypted REDCap database hosted on University of California San Francisco server [25].

### Statistical methods and data analysis

Physiologic and injury data from the CTR were used to calculate the KTS and RTS for each patient. The HEAIS value was the highest AIS value of any anatomic region for each patient. For example, if a patient received an AIS score of 6 (i.e., most severe) for an extremity and scores of 1 in every remaining anatomic region, the assigned HEAIS score would be 6. We defined scoring system feasibility as low rates of data missingness. To calculate feasibility, missing data resulting in incomplete scores for each scoring system were reported as percentages. Differences between groups with and without injury severity scores were compared via Pearson's chi-squared statistic, Pearson correlation coefficient, and 2x2 odds ratio (OR) tables. Median age differences between groups were compared by Wilcoxon rank-sum test. We used logistic regression to test for associations in disposition outcomes controlling for sex, injury severity (e.g., HEAIS), and injury mechanism. Additionally, logistic regression was used to test the associations between each injury severity score (RTS, KTS, HEAIS) and trauma mortality. Discrimination of hospital mortality for each injury scoring system was evaluated using receiver operating characteristic (ROC) curves. ROC curves for each scoring system were compared to evaluate score performance in the Cameroonian context. All statistical analyses were performed in Stata version 16 [26].

### Ethics

Approval to conduct this study was granted by the University of California, Los Angeles (IRB#19–000086) and University of California, San Francisco (IRB#13–12535) institutional review boards, as well as the Cameroon National Ethics Committee of the Division of Health Operations Research of the Ministry of Public Health (N˚2014/09/496/CE/CNERSH/SP). Patients were approached by Cameroonian research assistants for informed consent.

## Results

### Trauma population demographics

Over two years, 9,635 patients presented with injuries to the four hospitals and enrolled in the CTR. The median age was 30 (Interquartile range [IQR]: 22–40) years old and 69.9% of the cohort was male. RTIs were the largest portion of traumatic mechanisms (55.8%). Regarding disposition, 62.8% patients were discharged home from the ED and there were 206 (2.2%) in-hospital deaths (Table 1).

### Injury severity score missing data and feasibility

Data missingness varied by score type and showed HEAIS to be the most feasible score. A very small percentage of patients (2.5%) had missing HEAIS scores. In comparison, 66.9% (p = 0.04) of patients lacked data to calculate RTS while 75.2% (p<0.001) of patients did not have data to determine KTS (Table 2). The large majority of missing RTS (95.4%) and KTS (84.8%) data was attributed to missing respiratory rate values (Table 3). Out of the 206 deceased patients, 1.9% lacked HEAIS scores while 69.4% lacked RTS and 70.4% lacked KTS (p<0.001).

Data missingness varied by score and by site (Table 2) as site A lacked sufficient data to calculate HEAIS in 2.1% of patients, RTS in 1.3% of patients, and KTS in 2.8% of patients, respectively. Site B lacked HEAIS for 3.8% of patients yet could not calculate RTS for 54.6% and KTS for 61.3% of patients. Site C did not calculate HEAIS in 0.4% of patients and lacked data to calculate RTS and KTS in 99.1% of its patients. Site D had no HEAIS in 1.4% of patients and could not discern RTS and KTS 76.4% and 90.3% of its patients, respectively.

Among the population with HEAIS, 86.8% (n = 8361) had HEAIS data for all anatomic locations. In comparison to patients with HEAIS, patients with missing HEAIS experienced less RTIs (OR 0.74, p = 0.04). Patients without HEAIS were also less likely to be admitted to the ward (OR 0.48, p = 0.02). Female and younger patients were more likely to possess adequate data to calculate HEAIS rather than RTS or KTS (Table 4).

Compared to patients with RTS, patients with missing RTS data were younger (Pearson correlation coefficient -0.11, p<0.001), less likely to be male (OR 0.81, p<0.001), and more likely to have an urban residence (OR 1.42, p<0.001). Patients with missing RTS data were less likely to be victims of RTIs (OR 0.72, p<0.001) and more frequently suffered other traumatic mechanisms such as assaults and falls. Patients with missing RTS data demonstrated greater odds of discharge home from the ED (OR 1.89, p<0.001) and lower ward admission (OR 0.68, p<0.001) (Table 4).

Patients missing KTS data were younger (Pearson correlation coefficient -0.11, p<0.001), less likely to be male (OR 0.74, p<0.001), and more likely to live in an urban setting (OR 2.07, p<0.001) when compared to patients with KTS values. Patients with missing KTS data were less likely to experience RTIs (OR 0.70, p<0.001). Likewise, patients with missing KTS experienced greater odds of falls (OR 1.44, p<0.001) and assaults (OR 1.46, p<0.001). Regarding

**Table 1. Summary trauma population demographics (n = 9635).**

|  |  | Percentage (n) |
|---|---|---|
| Age (median, IQR) |  | 30 (22–40) |
| Sex |  |  |
|  | Male | 69.9 (6733) |
| Hospital |  |  |
|  | Site A | 4.9 (472) |
|  | Site B | 41.3 (3981) |
|  | Site C | 13.3 (1279) |
|  | Site D | 40.0 (3858) |
| Household area of residence (urban vs. rural) |  |  |
|  | Urban | 88.7 (8542) |
| Mechanism of injury |  |  |
|  | Road traffic injury | 55.8 (5373) |
|  | Assault | 13.6 (1306) |
|  | Fall | 13.4 (1288) |
|  | Stab/cut | 8.5 (817) |
|  | Other | 8.8 (851) |
| Heart rate on arrival in bpm (median, IQR) |  | 85 (75–96) |
| Systolic blood pressure on arrival in mmHg (median, IQR) |  | 126 (115–138) |
| Temperature on arrival in degrees Celsius (median, IQR) |  | 37.0 (36.5–37) |
| Respiratory rate on arrival in rpm (median, IQR) |  | 20 (18–24) |
| Abnormal vital signs* |  | 28.2 (2713) |
| Disposition |  |  |
|  | Discharged home | 62.8 (6055) |
|  | Left against medical advice | 13.3 (1284) |
|  | Admitted ward | 12.3 (1182) |
|  | Transferred | 4.8 (462) |
|  | Died | 2.1 (206) |
|  | Directly to operating room | 1.9 (187) |
|  | Admitted intensive care unit | 0.5 (47) |

n = number of patients; IQR = interquartile range; p-value = probability value; rpm = respirations per minute; bpm = beats per minute; mmHg = millimeters of mercury

*Abnormal vital signs were defined as respiratory rate greater than 20 rpm or less than 8 rpm, heart rate greater than 100 bpm or less than 60 bpm, systolic blood pressure less than 90 mmHg, or temperature less than 36 degrees Celsius

disposition, patients with missing KTS data had higher odds of discharge home from ED (OR 3.06, p<0.001) and lower odds of ward admission (OR 0.55, p<0.001) (Table 4).

Women presented with less severe injury compared to men (mean HEAIS 1.87 vs. 2.19, p<0.001). Women experienced more falls than men (16.8% vs. 12.1%, p<0.001), but men suffered more RTIs (57.1% vs. 54.6%, p<0.029) and stab wounds (10.3% vs. 4.4%, p<0.001) (S1 Table). Logistic regression revealed that women had greater odds of discharge home from the ED (S2 Table).

## Efficacy

Overall, 2,328 patients (24.2%) had sufficient data for all three severity scores. Each severity score was highly predictive of in-hospital mortality (p-value <0.001) (Fig 1). Comparing discrimination of mortality between scoring systems, RTS had an area under the receiver

**Table 2. Injury scoring data missingness by site (n = 9635).**

|  |  | Percentage (n) | p-value |
|---|---|---|---|
| **HEAIS** |  |  |  |
| Overall (n = 9635) |  | 2.5 (238) | Ref |
| By site |  |  |  |
|  | Site A (n = 472) | 2.1 (10) |  |
|  | Site B (n = 3981) | 3.8 (151) |  |
|  | Site C (n = 1279) | 0.4 (5) |  |
|  | Site D (n = 3858) | 1.4 (53) |  |
| **RTS** |  |  |  |
| Overall |  | 66.9 (6438) | 0.036* |
| By site |  |  |  |
|  | Site A | 1.3 (6) |  |
|  | Site B | 54.6 (2174) |  |
|  | Site C | 99.1 (1267) |  |
|  | Site D | 76.4 (2949) |  |
| **KTS** |  |  |  |
| Overall |  | 75.2 (7244) | <0.001* |
| By site |  |  |  |
|  | Site A | 2.8 (13) |  |
|  | Site B | 61.3 (2439) |  |
|  | Site C | 99.1 (1268) |  |
|  | Site D | 90.3 (3482) |  |

n = number of patients; p-value = probability value; HEAIS = Highest estimated abbreviated injury scale;

RTS = Revised Trauma Score; KTS = Kampala Trauma Score

* = statistically significant p-value (less that 0.05)

operating characteristics curve (AUC) of 0.95 (95% confidence interval [95%CI] 0.92–0.98), KTS had a 0.96 AUC (95%CI 0.93–0.99), and HEAIS had a 0.88 AUC (95%CI 0.84–0.92). Examination of the 9,269 patients with HEAIS scores demonstrated an AUC of 0.94 (95%CI 0.93–0.96).

**Table 3. Missing RTS and KTS data.**

|  |  | Percentage (n) |
|---|---|---|
| **Missing RTS score (n)** |  | 6438 |
|  | Missing RR | 95.4 (6145) |
|  | Missing SBP | 25.0 (1611) |
|  | Missing GCS | 1.6 (100) |
| **Missing KTS score (n)** |  | 7244 |
|  | Missing RR | 84.8 (6145) |
|  | Missing injury score | 27.4 (1985) |
|  | Missing SBP | 22.2 (1611) |
|  | Missing age | 2.0 (145) |
|  | Missing AVPU | 0.8 (57) |

RTS = Revised Trauma Score; KTS = Kampala Trauma Score; n = number of cases; GCS = Glasgow coma score;

SBP = systolic blood pressure; RR = respiratory rate; AVPU = alert, voice, pain, unresponsive

**Table 4. Patient demographics, injury characteristics, and outcomes by HEAIS, RTS, and KTS data missingness.**

| | | Missing HEAIS (OR 95%CI) | Missing RTS (OR 95% CI) | Missing KTS (OR 95% CI) |
|---|---|---|---|---|
| Age (Pearson coefficient) | | 0.01 | -0.11 ** | -0.11 ** |
| Sex | | | | |
| | Male | 1.03 (0.76–1.41) | 0.81 (0.74–0.89) ** | 0.74 (0.67–0.83) ** |
| Household | | | | |
| | Urban | 1.17 (0.73–1.96) | 1.42 (1.24–1.63) ** | 2.07 (1.81–2.38) ** |
| Mechanism | | | | |
| | Road traffic injury | 0.74 (0.55–0.99) * | 0.72 (0.66–0.79) ** | 0.70 (0.64–0.77) ** |
| | Assault | 0.74 (0.44–1.18) | 1.29 (1.13–1.47) ** | 1.46 (1.26–1.70) ** |
| | Fall | 0.97 (0.61–1.49) | 1.42 (1.25–1.63) ** | 1.44 (1.24–1.67) ** |
| | Stab/cut | 0.53 (0.24–1.03) | 1.03 (0.89–1.21) | 0.96 (0.81–1.14) |
| Disposition | | | | |
| | Discharged home | 1.40 (0.97–2.04) | 1.89 (1.73–2.07) ** | 3.06 (2.77–3.37) ** |
| | Left against medical advice | 1.06 (0.64–1.68) | 0.66 (0.58–0.74) ** | 0.49 (0.43–0.55) ** |
| | Admitted ward | 0.48 (0.22–0.91) * | 0.68 (0.60–0.77) ** | 0.55 (0.48–0.63) ** |
| | Transferred | 0.51 (0.14–1.35) | 0.37 (0.30–0.44) ** | 0.24 (0.20–0.29) ** |
| | Died | 1.20 (0.32–3.18) | 1.13 (0.83–1.55) | 0.78 (0.57–1.08) |
| | Directly to operating room | 1.32 (0.35–3.52) | 0.83 (0.61–1.13) | 0.60 (0.44–0.83) ** |
| | Admitted intensive care unit | 0 (0–4.90) | 0.80 (0.43–1.53) | 0.64 (0.34–1.26) |

OR = odds ratio; 95% CI = 95% confidence interval

* = p-value less than 0.05

** = p-value less than 0.001

## Discussion

Clinical assessment of injury severity using HEAIS strongly predicts hospital mortality and far exceeds conventional scoring systems in feasibility. When compared to scoring systems such as RTS and KTS, HEAIS has superior feasibility with slightly lower predictive capacity of mortality in a resource-limited setting. Though statistically inferior to KTS and RTS mortality predictions, the difference in mortality predictive capacity remains small and HEAIS demonstrates excellent overall discrimination of mortality. HEAIS scores were completed at a greater rate for young patients and female patients compared to RTS and KTS. Implementation of HEAIS may alleviate data disparities in historically underrepresented populations of trauma patients (i.e., women) and be a more equitable scoring system. While women are more likely to demonstrate data missingness in the calculation of RTS and KTS, it does not appear that women suffer worse outcomes than men. Additional data would be needed to determine if greater odds of discharge home correlate with worse health. Moreover, HEAIS is easier to calculate in fatally ill patients, as the majority of deceased patients had missing data precluding RTS and KTS calculation.

HEAIS is convenient to calculate as it solely relies on clinician findings of a head-to-toe physical exam, one of the core elements of a basic trauma assessment. As HEAIS does not rely on measurement of specific physiologic or radiologic data, routine estimation may be more achievable in diverse LMIC hospital settings. Ease of HEAIS implementation may allow for under-developed quality improvement systems to provide an early benchmark for injury severity. HEAIS without data for all anatomic locations would only affect results if the injury with the greatest severity was not logged in the CTR. Such an occurrence seems unlikely as it would constitute a major error in data collection among a regularly supervised, trained

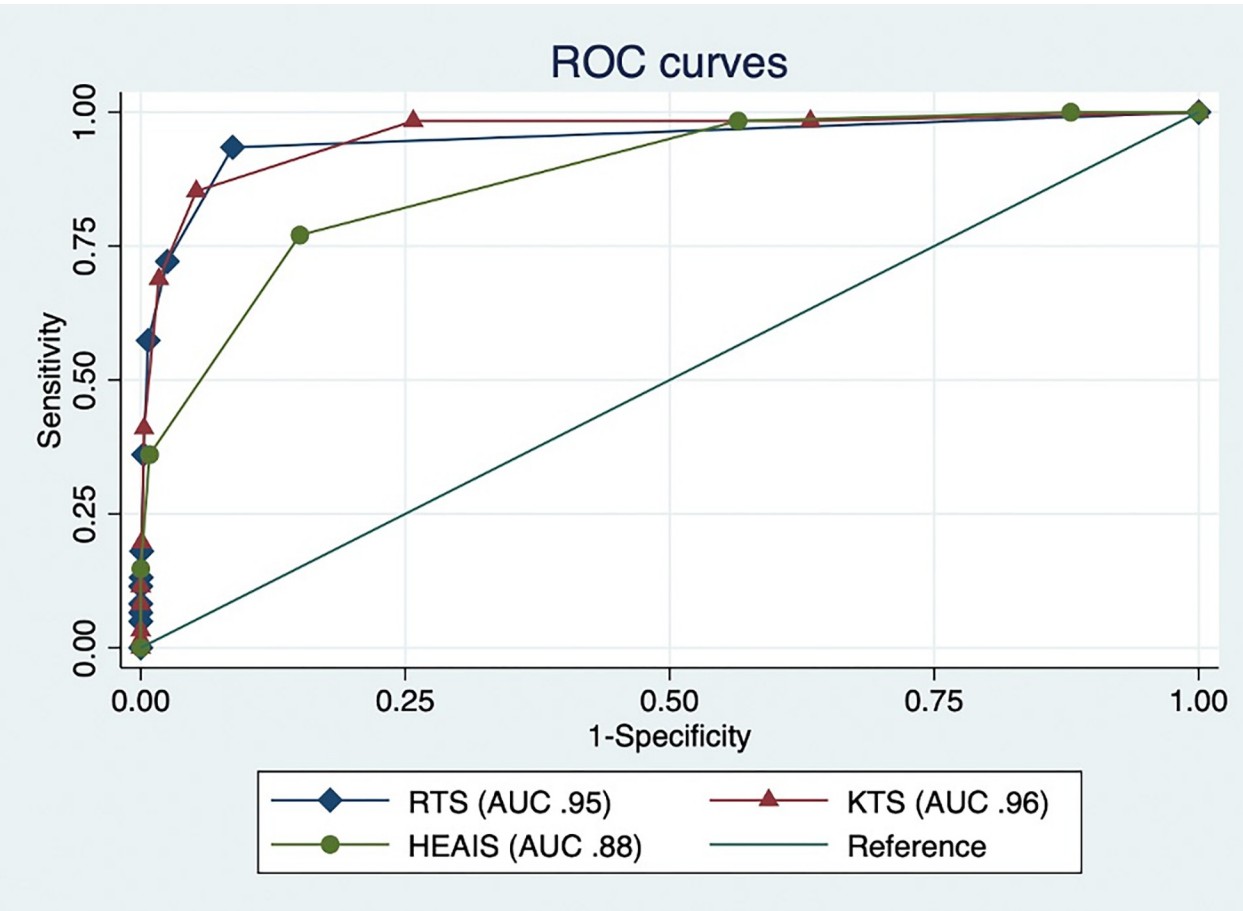

**Fig 1. Receiver operating characteristic curves.** Key: ROC = Receiver operating characteristic; AUC = Area under the ROC curve; 95% CI = 95% confidence interval; n = number of patients; HEAIS = Highest estimated abbreviated injury scale; RTS = Revised Trauma Score; KTS = Kampala Trauma Score.

research team. Among those without HEAIS for all anatomic locations, it is much more likely that data omission for an anatomic location occurred because findings in the location were unremarkable. Without HEAIS scores, over 6000 patients would not have an injury severity metric in this database. The much larger HEAIS cohort can provide risk adjustment in future QI initiatives and help predict in-hospital mortality going forward. Use of this novel injury scoring system improves quality of injury severity data and may facilitate trauma systems development.

The KTS was developed in a LMIC setting as an alternative to HIC injury severity scoring systems and has been validated in multiple settings over twenty years [27]. Moreover, RTS performs poorly in low-resource settings by consistently underestimating injury severity possibly due to observational variability between providers [9, 28–30]. In the CTR population, calculation of KTS and RTS score was greatly limited by absent respiratory rate values. It is unclear what barriers prevent the recording of respiratory rates as vital sign missingness remains common in LMIC trauma care [31]. Though 3% of CTR patients did not have vitals recorded due to the absence of working equipment, respiratory rate calculation does not require complex monitoring equipment. It is possible that respiratory rate is not recorded due to the large volume of clinical responsibilities placed on physicians. SSA physicians experience high rates of

burnout and excessive workload contributes to poor patient outcomes [32, 33]. The lack of respiratory rate collection reflects a systemic error exacerbated by shortages of trained workers and represents an opportunity for improvement with further training or task shifting intervention [34]. The improvement of respiratory rate data collection would allow for the calculation of KTS and RTS for thousands of CTR patients. While this improvement would increase the feasibility of KTS and RTS scores in the Cameroonian context, the scores remain less feasible than HEAIS.

Our findings bolster the limited existing literature describing predictive capacity of clinical judgement in LMICs. McLellan et al. calculated estimated injury severity scores (eISS) in a HIC patient population by having the trauma team leader estimate injury severity in the three most severely injured body systems [35]. The eISS correlated with traditional ISS calculated at time of death or discharge, highlighting the importance of clinical gestalt in initial trauma assessment. Similarly, a study of eISS in Cameroon correlated with traditional ISS and was an independent predictor of mortality at a threshold value [17]. Furthermore, a study in Ghana compared traditional KTS versus estimated "physician opinion KTS" in trauma patients [16]. The study obtained physician opinion KTS by combining vital sign measurements with ED provider estimation of the number of serious injuries present upon arrival. Physician opinion KTS greatly simplified the KTS calculation and increased feasibility, but still required measurement of physiologic parameters, which can be a limitation in a different, low resource setting [16].

It is important to emphasize there likely will never be a single trauma scoring system functional in all low-resource settings. We concur with other studies that caution that the choice of an injury scoring system requires intense consideration of local resources and needs [10, 15]. Simplification of the injury severity estimation process is especially important in settings where staffing resources are scarce. While HIC facilities often have multiple full-time staff members dedicated to measuring trauma outcomes, overworked clinicians in short-staffed LMIC facilities often do not have time to compile complicated severity scoring metrics.

In this study, we present a real-world evaluation of feasibility and efficacy of clinician estimated severity in a large, prospective trauma LMIC cohort. However, there are several notable study limitations. First, our study only measured outcomes for those who presented to the hospital and underrepresented the true trauma mortality rate [18, 19]. Trauma patients who did not present to the ED, did not receive prehospital transport, died at the scene, or left the hospital AMA and died afterwards were not included in this calculation. Prior Cameroonian community-based surveys reveal that about 60% of the injured population presents for formal care [20]. Thus, our HEAIS findings cannot be generalized to the population cohort that does not present to the hospital for formal evaluation. Discrimination of differences between the complete HEAIS and missing HEAIS groups was limited in power due to small sample. Furthermore, HEAIS relies on individual clinician exam and experience. It is possible that providers with differing clinical backgrounds may not reproduce similar results due to differing perspectives of injury estimation. Additional data is needed to evaluate the influence of provider characteristics and training on HEAIS discrimination of mortality. Finally, LMICs are diverse, and generalizability will need to be validated in other LMIC contexts. The CTR is currently expanding from four to ten facilities across Cameroon, and discrimination of HEAIS will continue to be evaluated in this broader application.

## Conclusions

In resource-limited contexts where traditional scoring systems are not feasible, utilization of clinical gestalt metrics like HEAIS could facilitate improved capture of severity data and expand access and strengthen quality improvement programming.

## Supporting information

**S1 Table. Injury severity, mechanism, and disposition outcomes by sex.**
(DOCX)

**S2 Table. Logistic regression of disposition outcomes.**
(DOCX)

**S1 Data. De-identified data.**
(XLSX)

## Acknowledgments

We acknowledge the dedication and effort of the Cameroonian trauma registry staff at each participating hospital who are involved in data entry and the support of the Cameroonian Ministry of Public Health.

## Author Contributions

**Conceptualization:** Rasheedat Oke, S. Ariane Christie.

**Formal analysis:** Mark T. Yost, S. Ariane Christie.

**Funding acquisition:** Catherine Juillard.

**Project administration:** Melissa M. Carvalho, Lidwine Mbuh, Fanny N. Dissak-Delon, Rasheedat Oke, Debora Guidam, Rene M. Nlong, Mbengawoh M. Zikirou, David Mekolo, Louis H. Banaken, Catherine Juillard, Alain Chichom-Mefire.

**Supervision:** Catherine Juillard, Alain Chichom-Mefire, S. Ariane Christie.

**Writing – original draft:** Mark T. Yost.

**Writing – review & editing:** Melissa M. Carvalho, Rasheedat Oke, Catherine Juillard, S. Ariane Christie.

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
