## [Decision Letter · Decision Letter 0]

1 Feb 2023

PGPH-D-22-01916

Back to the Basics: Clinical assessment yields robust mortality prediction and increased feasibility in low resource settings

Dear Dr. Yost,

Thank you for submitting your manuscript to PLOS Global Public Health. After careful consideration, we feel that it has merit but does not fully meet PLOS Global Public Health’s publication criteria as it currently stands. Therefore, we invite you to submit a revised version of the manuscript that addresses the points raised during the review process.

We look forward to receiving your revised manuscript.

Kind regards,

Jagnoor Jagnoor

Academic Editor

Journal Requirements:

1. Please send a completed 'Competing Interests' statement, including any COIs declared by your co-authors. If you have no competing interests to declare, please state "The authors have declared that no competing interests exist". 

3. In the online submission form, you indicated that "Data available on reasonable request". All PLOS journals now require all data underlying the findings described in their manuscript to be freely available to other researchers, either 1. In a public repository, 2. Within the manuscript itself, or 3. Uploaded as supplementary information.

4. Please upload all main figures as separate Figure files in .tif or .eps format only. For more information about how to convert and format your figure files please see our guidelines:

Additional Editor Comments (if provided):

Reviewers' comments:

Reviewer's Responses to Questions

**Comments to the Author**

1. Does this manuscript meet PLOS Global Public Health’s publication criteria? Is the manuscript technically sound, and do the data support the conclusions? The manuscript must describe methodologically and ethically rigorous research with conclusions that are appropriately drawn based on the data presented.

Reviewer #1: Yes

Reviewer #2: Yes

2. Has the statistical analysis been performed appropriately and rigorously?

Reviewer #1: Yes

Reviewer #2: Yes

3. Have the authors made all data underlying the findings in their manuscript fully available (please refer to the Data Availability Statement at the start of the manuscript PDF file)?

Reviewer #1: Yes

Reviewer #2: Yes

4. Is the manuscript presented in an intelligible fashion and written in standard English?

Reviewer #1: Yes

Reviewer #2: Yes

5. Review Comments to the Author

Reviewer #1: The authors report a well conducted study to estimate injury severity and predict mortality in resource-constrained settings more easily. The use of the physical-exam based Highest Estimated Abbreviated Injury Score (HEAIS) has great promise in triaging patients and managing hospital resources. While the study and results are very well presented, I have a few questions for the authors:

1. The results indicate that HEAIS is statistically inferior to KTS and RTS but with good overall discrimination of mortality. It is therefore concerning that the authors conclude in the discussion section that HEAIS is “comparable” to KTS and RTS. This may give the erroneous impression that HEAIS is equivalent to KTS and RTS in predicting mortality. Could the authors address this?

2. It would be good for the authors to clarify if patients with HEAIS scores had data for ALL anatomic locations? If this was not the case, an explanation of how this may or may not affect the results would be helpful

3. The authors clearly identified missing “Respiratory Rate” as main reason for not being able to calculate KTS or RTS. It would be good for the authors to comment in the discussion section on the possibility of HEAIS + RR in increasing injury severity predictiveness in resource-constrained settings.

Congratulations once again for a well-executed study.

Reviewer #2: 1- Manuscript addresses the important public health problem in sub-Saharan Africa (SSA) of injury related death and disability. The tool is feasible and was tested in several hospitals with different characteristics and catchment areas.

2- The authors state that “HEAIS scores were completed at a greater rate for young patients and female patients compared to RTS and KTS”. This is not well described in the result section. It is described as ORs in table 4 but should be mentioned clearly in a sentence or two in the result section

3- In the methods section, hospitals are described as first, second, third and final facility, but described in the tables as site A, B, C, D. I suggest using the same description in both the method and the result section to decrease confusion

4- The scale being tested required less use of technology, was more feasible and may alleviate data disparities in women as trauma patients. Are the authors able to include more details as to the types or setting of injury? Do women experience different patterns of injury or are more demanding scales not used for women because women get less quality services compared to men?

5- Description of local teams in each hospital as well as the local team of investigators is not very clear, were there local PIs at each site? I suggest acknowledging local teams who are not part of the authors.

I commend the authors for addressing trauma in SSA, creating the registry and discussing very relevant aspects of care such as workload and its impacts of assessment of patients and in pursing quality improvement

6. PLOS authors have the option to publish the peer review history of their article (what does this mean?). If published, this will include your full peer review and any attached files.

**Do you want your identity to be public for this peer review?** For information about this choice, including consent withdrawal, please see our Privacy Policy.

Reviewer #1: No

Reviewer #2: **Yes: **Nazik Hammad

---

## [Editor Report · Decision Letter 1]

6 Mar 2023

Back to the Basics: Clinical assessment yields robust mortality prediction and increased feasibility in low resource settings

PGPH-D-22-01916R1

Dear Dr. Yost,

We are pleased to inform you that your manuscript 'Back to the Basics: Clinical assessment yields robust mortality prediction and increased feasibility in low resource settings' has been provisionally accepted for publication in PLOS Global Public Health.

Best regards,

Jagnoor Jagnoor

Academic Editor